# Non-Lactic Probiotic Beverage Enriched with Microencapsulated Red Propolis: Microorganism Viability, Physicochemical Characteristics, and Sensory Perception

**Iara Ferreira [1], Dirceu de Sousa Melo [1], Marly Silveira Santos [2], Disney Ribeiro Dias [3], Carolina Oliveira de Souza [4], Carmen Sílvia Favaro-Trindade [5], Lorena Silva Pinho [5], Rogeria Comastri de Castro Almeida [2], Karina Teixeira Magalhães-Guedes [4] and Rosane Freitas Schwan [1,*]**

[1] Post-Graduate Program in Agricultural Microbiology, Department of Biology, Federal University of Lavras, Lavras 37200-900, MG, Brazil

[2] Food Science Department, School Nutrition Federal University of the Bahia (UFBA), Rua Basílio da Gama, s/n—Campus Canela, Salvador 40110-907, BA, Brazil

[3] Post Graduate Program in Foos Science, Department of Food Science, Federal University of Lavras, Lavras 37200-900, MG, Brazil

[4] Post-Graduate Program in Food Science, Federal University of Bahia (UFBA), Rua Barão de Jeremoabo, 147–Campus Ondina, Salvador 40170-115, BA, Brazil

[5] Faculty of Animal Science and Food Engineering, University of São Paulo, Av. Duque de Caxias Norte, 225, CP 23, Pirassununga 13535-900, SP, Brazil

* Correspondence: rschwan@ufla.br; Tel.: +55-35-38291614

**Abstract:** This work aimed to develop a non-dairy functional beverage fermented with probiotic strains and fortified with Brazilian red propolis (microencapsulated and extracted). The non-dairy matrix consisted of oats (75 g), sunflower seeds (175 g), and almonds (75 g). It was fermented by a starter co-culture composed of *Lactiplantibacillus plantarum* CCMA 0743 and *Debaryomyces hansenii* CCMA 176. Scanning electron microscopy analysis was initially performed to verify the integrity of the microcapsules. The viability of the microorganisms after fermentation and storage, chemical composition (high performance liquid chromatography (HPLC) and gas chromatography coupled to mass spectrometry (GC-MS) analyses), rheology, antioxidant activity, and sensory profile of the beverages were determined. After fermentation and storage, the starter cultures were well adapted to the substrate, reducing the pH (6.50 to 4) and cell count above 7.0 log CFU/mL. Lactic acid was the main organic acid produced during fermentation and storage. In addition, 39 volatile compounds were detected by gas chromatography coupled to mass spectrometry (GC-MS), including acids, alcohols, aldehydes, alkanes, alkenes, esters, ethers, phenols, terpenes, and others. The addition of propolis extract increased the antioxidant and phenolic activity and the presence of volatile esters but reduced the beverage's acceptability. The addition of microencapsulated propolis was more associated with the presence of higher alcohols and had similar acceptance to the control beverage. The combination of a non-dairy substrate, a starter co-culture, and the addition of propolis led to the development of a probiotic beverage with great potential for health benefits.

**Keywords:** yeasts; plant matrix; volatiles; antioxidants; co-culture

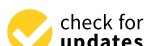



## 1. Introduction

Functional foods offer several health benefits in addition to the nutritional value inherent to their chemical composition [1]. Currently, consumers are increasingly concerned about consuming products with characteristics that play an important role in disease prevention and health promotion. In this context, interest in consuming functional foods has grown [2–4]. Among functional foods, beverages are the fastest-growing segment in the functional food market [5]. In the U.S., functional beverages accounted for about 59% of the functional food market in 2012. By 2025, these beverages are expected to account for

40% of the total consumer demand [5]. Products containing probiotics emerge as one of the main functional foods consumers prefer due to their health benefits [2,3,6].

When administered adequately, probiotic products contain live microorganisms that benefit the host's health [3,7,8]. The global market for probiotics is increasing significantly and is expected to reach USD 65.87 billion by 2024 with 69% of this market value coming from non-dairy products [9,10]. For a long time, dairy products have been used to deliver probiotic microorganisms [11]. However, the increasing emergence of lactose intolerance, milk allergies, and the prevalence of hypercholesterolemia along with the growth of vegetarian habits are leading to a growing demand for non-dairy alternatives [2,9,12–14].

Oats, almonds and sunflower seeds are promising non-dairy substrates for plant "milk" production. Oats are cereals rich in β-glucan, dietary lipids, proteins, starch, antioxidant phenolic compounds, and soluble fibers such as oligosaccharides and polysaccharides [15,16]. Almonds are an excellent source of lipids and proteins, rich in fatty acids, vitamins, and minerals [17], have prebiotic properties [15], and are considered beneficial for intestinal transit in addition to acting in the prevention of anemia, cancer, and also in the protection against free radicals [18]. Sunflower seeds are sources of potassium, phosphorus, calcium, and contain essential amino acids, such as valine, isoleucine, and leucine [19].

Combining probiotic microorganisms and food sources with bioactive properties can make a product even more functional, expanding the health benefits. In this context, Brazilian red propolis (B.R.P.) is a potential additive for functional food products. B.R.P. is found on the northeastern Brazilian coast, produced by *Apis mellifera* bees in association with the *Dalbergia ecastophyllum* plant [20]. B.R.P. is known for its anti-inflammatory, anti-allergic, anticancer [21], antioxidant [22], antibacterial [23], antifungal [24], and antiviral activity [25]. However, incorporating propolis into food is a challenge as it has a low solubility in water and strongly alters the sensory characteristics due to its solid and unpleasant taste and odor, compromising the acceptability of the product [26]. Technological strategies such as encapsulation can minimize the sensory impacts of B.R.P. upon inclusion in foods. Encapsulation is a process that traps a substance (active agent) in a wall material producing micro- or nanoparticles [27].

Considering all the aspects, the objective of the present study was to develop a fermented non-dairy beverage using strains with probiotic attributes enriched with B.R.P. (microencapsulated and extract) to obtain a product with more significant functional properties. The following were evaluated: the fermentation process, the viability of microorganisms during refrigerated storage, antioxidant and rheological properties, and the sensory profile of the beverage. Analyses by high performance liquid chromatography (HPLC) and gas chromatography coupled to mass spectrometry (GC-MS) were also performed in this study. HPLC and GC-MS techniques are important to chemically characterize the studied product [1,28]. Volatile, non-volatile, nutritional, and functional compounds can be detected and quantified using these techniques [1,28].

## 2. Materials and Methods

### 2.1. Production of the Microcapsules

The red propolis microencapsulated and red propolis extract were provided by the University of São Paulo (U.S.P.), Faculty of Animal Science and Food Engineering. Red propolis (RdProp) is a resin produced by *Apis mellifera* bees, which collect the reddish exudate on the surface of its botanic source, the species *Dalbergia ecastophyllum*, popularly known in Brazil as "rabo de bugio". Considered as the 13th type of Brazilian propolis, this resin has been gaining prominence due to its natural composition, rich in bioactive substances. Their properties come from countless compounds, including terpenes, pterocarpans, prenylated benzophenones, and especially flavonoids. This last compound class has been indicated as the compound responsible for its potent pharmacological actions, highlighting the antimicrobial, anti-inflammatory, antioxidant, healing, and antiproliferative activities [28].

Arabic gum solution (20%) was used as a carrier to prepare microcapsules. An individual solution of the carrier (Arabic gum 20%) was prepared at the concentration

of 20 g/100 mL. A ratio of propolis extract to carrier solution equal to 1:6 (*w/w*) was also employed [28]. A total of 100 mL of red propolis extract was used in this study. The carrier agent solution and the propolis extract were dispersed with a homogenizer (Ultra-Turrax T25; I.K.A., Königswinter, Germany) for 2 min at 15,000 rpm. The formulation was then atomized with a spray dryer (model MSD 1.0, Labmaq, Brazil). The operational conditions of the spray dryer were as follows: inlet temperature of 120 °C; outlet temperature of 91 °C; air flow of 0.60 $m^3$/min; feed flow of 0.60 $m^3$/min; and nozzle diameter of 1.3 mm [28]. At the end of each drying session, the powders were gathered, placed in closed vials covered with aluminum foil, and kept at room temperature in a dry and dark place until analysis. The encapsulation process and all analyses were conducted in triplicate.

### 2.1.1. Moisture

The moisture of microencapsulated propolis was determined by the gravimetric method. Approximately 3 g of the samples were weighed into porcelain capsules of known mass. The capsules were yeasted in an oven at 105 °C until constant weight according to the methodology of [29]. The moisture was calculated according to Equation (1):

$$\text{Moisture} = \frac{\text{mi} - \text{mf}}{\text{mi}} \tag{1}$$

where mi is the initial mass, and mf is the final mass.

### 2.1.2. Morphology and Average Particle Diameter

The morphology was evaluated using a scanning electron microscope (S.E.M.) (model Jeol JSM 6360 LV, Jeol EUA Inc, Peabody, MA, USA) with an accelerating voltage of 20 kV. The analysis was performed under a high vacuum and without the application of tilt. The microcapsules were mounted on aluminum bases (stub) adhered by double-sided carbon adhesive tape. They were then coated with gold (3 nm) with a vacuum spray applicator [30]. The average diameter of the microcapsules was evaluated using ImageJ 2014 software (Rasban, National Institute of Health, Bethesda, MD, USA) [31].

### 2.2. Determination of the Minimum Inhibitory Concentration (M.I.C.)

The M.I.C. test was performed according to Gonçalves et al. [32] with some modifications to determine the concentration of red propolis to be used in the work that would not affect microorganisms during fermentation. For this, microdilution tests [33] were performed in 96-well microplates. The M.I.C. was determined using Y.P.D. broth for yeast and M.R.S. broth for bacteria. In each well, 100 μL of red propolis (microencapsulated and extract) solubilized in the microorganism's growth medium at different concentrations (10,000 μg/mL, 5000 μg/mL, 2500 μg/mL, 1250 μg/mL, 625 μg/ mL, 312.5 μg/mL, 156.25 μg/mL, and 78.125 μg/mL) and 10 μL of test microorganism suspension were added together with 100 μL of stereo culture medium; 200 μL of culture medium with 10 μL of the test microorganism were used as a positive control and 210 μL of culture medium without the microorganism as a negative control. The microplates were incubated at 30 °C for yeast and 37 °C for lactic acid bacteria (L.A.B.) for 24 h. After this process, 10 μL was removed from each well of the polystyrene microplate and plated using the micro drop technique. The plates were incubated under the same conditions mentioned above. The M.I.C. was defined as the concentration of propolis (extract and microencapsulated) that did not affect microbial growth. The experiment was performed in triplicate for each strain.

### 2.3. Microorganisms and Culture Conditions

The bacteria *Lactiplantibacillus plantarum* (formerly *Lactobacillus plantarum*) CCMA 0743 and the yeast *Debaryomyces hansenii* CCMA 1761, provided by the Agricultural Microbiology Culture Collection (CCMA) of the Federal University of Lavras (Brazil), were used in this work. Both microorganisms were previously selected based on their probiotic technological characteristics [34,35]. Yeast and L.A.B. were stored at −80 °C in yeast extract–peptone

dextrose (Y.P.D.) broth at 10 g/L of yeast extract (Merck, Darmstadt, Germany), 10 g/L of peptone (Himedia, Mumbai, India), 20 g/L of glucose (Merck, Darmstadt, Germany), and Man, Rogosa, and Sharpe broth (M.R.S.) (Merck, Darmstadt, Germany), respectively, with 20% (*v/v*) glycerol. The yeast strain and L.A.B. were reactivated on Y.P.D. agar and M.R.S. agar, respectively, and incubated for 24 h at 30 °C for yeast and 37 °C for L.A.B.

### 2.4. Beverage Preparation

The medium for the controlled fermentation tests was prepared according to Ferreira et al. [36]. The non-dairy matrix (N-DM) consisted of oats (75 g), sunflower seeds (175 g), and almonds (75 g) in a ratio of 2:1:1, respectively. After the pasteurization process, the microorganisms were added according to item 2.5. The beverage was fermented for 24 h at 30 °C. At the end of fermentation, propolis (microencapsulated and extract) was added and allowed to ferment for another 24 h, totaling 48 h of fermentation. Substrates were purchased at the local market in Lavras, Minas Gerais, Brazil.

### 2.5. Fermentation and Sampling Tests

Three treatments were performed from the non-dairy matrix (N-DM) consisting of oats (75 g), sunflower seeds (175 g), and almonds (75 g) in the ratio of 2:1:1, respectively: 1 Control—consists only of the fermented beverage (N-DM); 2 Extract—beverage (N-DM) with propolis extract (DPE); 3 Microcapsule—beverage (N-DM) with microencapsulated propolis (D.M.P.).

Yeast and L.A.B., already grown, were centrifuged and resuspended in the fermentation medium with a population of 6.0 log CFU/mL for yeast and 7.0 log CFU/mL for bacteria. The fermentations were carried out in 500 mL Erlenmeyer flasks containing 300 mL of the vegetable beverage at 30 °C for 48 h. All assays were performed in triplicate. Samples were taken at 0 and 24 h of fermentation to count the microorganisms. After removing the samples within 24 h of fermentation, propolis extract (2500 μg/mL) was added for the DPE treatment, while for the D.M.P. treatment, microencapsulated propolis was added at the same concentration. All treatments were fermented for another 24 h. At the end of fermentation, the beverages were refrigerated at 4 °C for 28 days. Samples at 7, 14, 21, and 28 days of storage were also taken for subsequent analysis.

### 2.6. Enumeration of Microorganisms

The total populations of L.A.B., Enterobacteriaceae, and filamentous fungi were determined by plating on M.R.S. agar (supplemented with 4 mL/L of nystatin), red-violet bile glucose (VRBG) agar (Merck) medium, and Dichloran Rose Bengal Chloramphenicol agar medium (DRBC) (Kasvi), respectively. Plates were incubated at 37 °C (L.A.B. and Enterobacteriaceae) and 30 °C (filamentous fungi) for 24 h and 7 days for filamentous fungi. After the incubation period, the colony-forming units (CFU) were enumerated. Analyses were performed in triplicate.

### 2.7. Analysis of Organic Acids, Alcohols, and Carbohydrates

High-performance liquid chromatography (HPLC) was used to evaluate the samples' organic acids, alcohols, and carbohydrates at 0 h, 48 h of fermentation, 14 days, and 28 days of storage. The samples were centrifuged (10,000 rpm) for 5 min at 4 °C. The supernatant was recovered and filtered with sterile syringe filters (0.22 μm pore size; Kasvi, Brazil). Perchloric acid (1 μL) was added to the acid samples only to equalize the pH of the sample with that of the mobile phase (pH 2.1) followed by centrifugation and filtration as described above. Analyses were performed using an HPLC (model LC-10Ai; Shimadzu Corp., Tokyo, Japan) equipped with a dual-detection system consisting of a UV-vis detector (SPD-10Ai; Shimadzu) and a U.V. index detector refraction (RID-10Ai; Shimadzu). A Shimadzu ion exclusion column (Shim-pack SCR-101H, 7.9 mm × 30 cm) was used for alcohol, carbohydrate (30 °C), and organic acid (50 °C) determination [37]. The mobile phase was ultrapure water (carbohydrates and alcohols) and acidified ultrapure water

(pH 2.1) for acids at a flow rate of 0.6 mL/min. The compounds were identified based on the retention time of the standards, and their concentrations were determined by the external calibration method. All samples were examined in triplicate.

### 2.8. Extraction of Volatile Compounds and Gas Chromatography with a Mass Spectrometer (GC-MS)

The headspace solid-phase microextraction technique (SPME-HS) was used to extract volatile compounds from the samples, as described by Menezes et al. [38] with minor modifications. A total of 2 mL of each sample was placed in 15 mL vials. A 50/30 μm divinylbenzene/carboxene/polydimethylsiloxane (DVB/CAR/PDMS) fiber supplied by Supelco (Bellefonte, PA, USA) was used to extract the volatile compounds. This fiber was balanced for 15 min at 60 °C and then exposed to the samples for 30 min at the same temperature. Injections were performed by fiber exposure for 2 min.

Volatile compounds were analyzed by gas chromatography–mass spectrometry (GC-MS) (Model GCMS-QP2010SE; Shimadzu, Tokyo, Japan) equipped with a Carbowax column (30 m × 0.25 mm id × 0.25 μm film thickness). The oven temperature was set to 40 °C for 5 min and increased until it reached 220 °C (at a rate of 10 °C/min) and was finally maintained at this temperature for 10 min. The carrier gas was high-purity helium at 0.7 mL/min. Splitless injection was used. The mass-selective detector was a quadrupole with an electronic impact ionization system at 70 eV and 250 °C. Volatile compounds were identified, and peak areas were used for relative quantification using the GC/MS Solution software (version 2.6). Volatile compounds were identified by comparing mass spectra with the NIST11 library. Furthermore, a series of alkanes (C7–C19) was used to calculate each compound's retention index (R.I.) and compare them with R.I. values from academic literature.

### 2.9. Antioxidant Activity and Total Polyphenols

The extracts were obtained according to the method described by De Souza et al. [39] with minor modifications. Briefly, 5 mL of the beverage samples were added into centrifuge tubes and extracted sequentially with 10 mL of 50% (*v/v*) methanol at 25 °C for 1 h. The tubes were centrifuged at 7000 rpm at room temperature for 10 min, and the supernatant was recovered. Then, 10 mL of 70% (*v/v*) acetone was added to this supernatant and recovered at room temperature. Samples were extracted for 1 h and centrifuged under the above conditions. Extracts composed of methanol/acetone were used to determine the antioxidant activity.

#### 2.9.1. Phosphomolybdenum Complex Method (P.C.M.)

The antioxidant activity was determined by P.C.M. according to the modified methodology described by Prieto, Pineda, and Aguilar [40]. A 100 μL aliquot of the sample extract was placed in tubes and mixed with 3 mL of reagent solution (1.8 M sulfuric acid, 28 mM sodium phosphate, and 4 mM ammonium molybdate). The tubes were capped and incubated in a water bath at 95 °C for 90 min. Then, the samples were cooled to room temperature, and the absorbance of the phosphomolybdenum complex was measured at 695 nm. A mixture containing 50% methanol and 70% acetone (1:1) was used as a blank. Quantification was based on a standard curve of ascorbic acid (1.95 to 500 μg), and results were expressed in mg of ascorbic acid equivalents (A.A.E.) per mL of sample.

#### 2.9.2. Elimination of the ABTS Radical

The radical-scavenging activity of the fermented beverage samples was evaluated using the radical 2,20-azinobis-3-ethylbenzothiazoline-6-sulfonate (ABTS) according to Re et al. [41] with minor modifications. Briefly, 30 μL of sample extract or reference substance (Trolox) was added to 3 mL of ABTS radical solution and allowed to react in the dark for 6 min. The absorbance was measured at 734 nm. Quantification was based on a standard

Trolox curve, and results are expressed as micromoles of Trolox equivalents (T.E.) per mL of sample ($\mu$mol T.E./mL).

### 2.9.3. Determination of Total Polyphenol Content (TPC)

The total polyphenol content (TPC) was determined by a spectrophotometric assay (UV-VIS Spectrum SP-2000 UV, Biosystems) following the Follin–Ciocalteau methodology [42]. In summary, 0.5 mL of sample extract, 2.5 mL of Folin–Ciocalteau reagent (10%), and 2.0 mL of $Na_2CO_3$ (4% *w/v*) were homogenized and incubated at room temperature (25 °C) in the dark for 2 h. The absorbance of the samples was measured at 750 nm. Phenolic concentrations were calculated based on the standard curve of gallic acid (ranging from 10 to 100 $\mu$g/mL) and are expressed in milligrams of gallic acid equivalents per mL of sample (mg G.A.E./mL). All analyses were performed in triplicate.

### 2.10. Rheological Behavior of Beverages

The rheological behavior of the beverages was performed at 4 °C using a rotational viscometer (Brookfield, DV III Ultra, Brookfield Engineering Laboratories, Stoughton, MA, USA) coupled to a thermostatic bath (Brookfield, EX 200) for sample temperature control. For sample control, the coaxial SC4–18 shear sensor was used [43]. The tests were performed with three replications for each sample.

### 2.11. Sensory Analysis

Sensory analysis was performed after approval by the Ethics Committee of the Federal University of Lavras (CAAE: 48881621.0.0000.5148). The consumer acceptance test was carried out according to the hedonic scale of nine categories ranging from disliked very much (1) to liked very much (9). Consumers rated the samples for appearance, texture, taste, and overall impression. Together, the purchase intention test was performed according to a 5-point hedonic scale ranging from I would not buy (1) to I would buy (5) [44].

The tests were carried out in closed cabins with white lighting at the Sensory Analysis Laboratory of the Federal University of Lavras (Lavras, Minas Gerais, Brazil). Samples were labeled with three random digits on a white surface. One hundred and eight untrained panelists participated in the analysis; they were between 18 and 55 years old and were students and workers at the Federal University of Lavras. Random samples of 15 mL were served at a consumption temperature of 4.0 ± 1.0 °C. Mineral water was provided for mouthwash during sample evaluation.

### 2.12. Statistical Analysis

The experiment was carried out in a completely randomized design (D.I.C.). Data were submitted to analysis of variance (ANOVA, Statsoft, OK, USA), and means were compared using the Scott–Knott test with significance at $p < 0.05$. Data were analyzed using the SISVAR software version 5.8.

Volatile compounds were analyzed by principal component analysis (P.C.A.) using SensoMaker 1.92 software.

For sensory analysis, ANOVA was performed followed by Tukey's test to calculate significant differences with $p < 0.05$. Analyses were performed using Statistica software version 10.0 (Statsoft, Tulsa, OK, USA).

## 3. Results and Discussions

### 3.1. Moisture, Morphology, and Particle Diameter

After the microencapsulation process, it was important to check the material's water content and morphology as it interferes with the microcapsule's stability and efficiency.

Low moisture is required to avoid the agglomeration of the particles, which can result in hardening of the powders, hampering the flow and dispersion [28]. The moisture of the propolis microparticles was 9%; the values were similar to the results obtained by

Andrade et al. [45] who reported values ranging from 5.84% to 9.96% when propolis was sprayed with maltodextrin and Arabic gum as carriers.

Figure 1 shows the S.E.M. microphotographs of the microencapsulated propolis. The microphotographs show morphological similarity, with irregular shapes and sizes, some with a smooth surface and others with a rough surface with small concavities, which can be attributed to the rapid evaporation of liquid droplets during the spray-drying process [46]. A typical feature of particles produced by spray drying, which can also be observed in the images, is the formation of agglomerates where small particles are positioned around larger particles. Busch et al. also reported similar morphological features [47] in their studies on propolis microparticles using Arabic gum and maltodextrin as encapsulating materials. According to Tonon et al. [48], particles with rough surfaces have more extensive contact areas than those with smooth surfaces, which makes them more susceptible to degradation reactions such as oxidation. Therefore, the microparticles should have a uniform surface with minimal cracks, irregularities, or roughness as it promotes better-encapsulated material functionality [28,49].

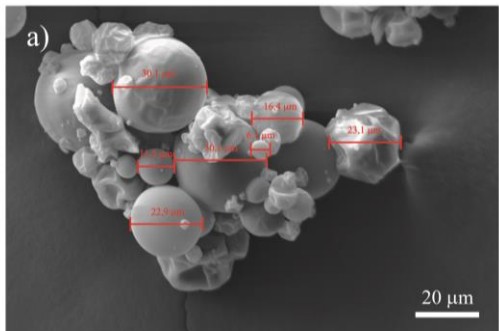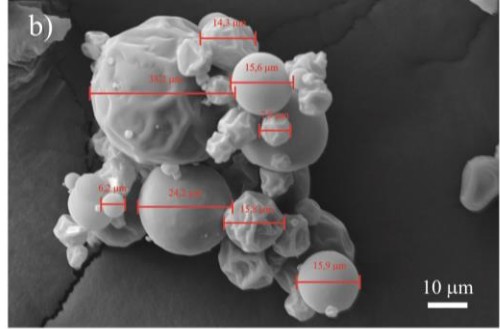

**Figure 1.** Scanning electron micrograph of microencapsulated propolis at different magnifications and particle diameters. (**a**) 1.57 K increase and (**b**) 1.94 K increase.

The microparticles showed particle sizes ranging from 6.1 to 38.2 μm under S.E.M. observation; the product's particle size is in the microencapsulation size requirement range of 3 to 800 μm [50]. Therefore, the propolis microencapsulation process was efficient. The size of the microparticles is important because it affects physical and functional properties such as solubility and hygroscopicity, as these depend on the contact surface [51].

*3.2. Microbial Growth and Beverage Acidification*

Figure 2 shows the growth profile of the microorganisms during fermentation. The growth of both strains (*D. hansenii* CCMA 1761 and *L. plantarum* CCMA 0743) was monitored along the fermentation process. After 24 h, the *D. hansenii* population demonstrated above 7.0 log CFU/mL (Figure 2a) and maintained these counts after the addition of propolis (48 h). The D.M.P. assay expressed growth of 7.51 log CFU/mL (8.68%), while the DPE and control showed values around 7.41 log CFU/mL (9.77%) and 7.49 log CFU /mL (9.02%), respectively, with no significant difference between treatments ($p > 0.05$). Similar behavior was observed for L.A.B. (Figure 2b). After 48 h of fermentation (24 h after the addition of propolis), the counts of all treatments showed values above 8 log CFU/mL. D.M.P. exhibited values around 8.55 log CFU/mL, while the DPE and control presented counts of 8.47 log CFU/mL and 8.51 log CFU/mL, respectively. Alemneh, Emire, and Hitzmann [12], working with a functional probiotic beverage based on cereals (teff) fermented by *lactobacillus* strains, also observed bacterial growth above 8 log CFU/mL during fermentation. The results reveal that the addition of propolis (microencapsulated or extract) did not interfere negatively with the growth of the microorganisms.

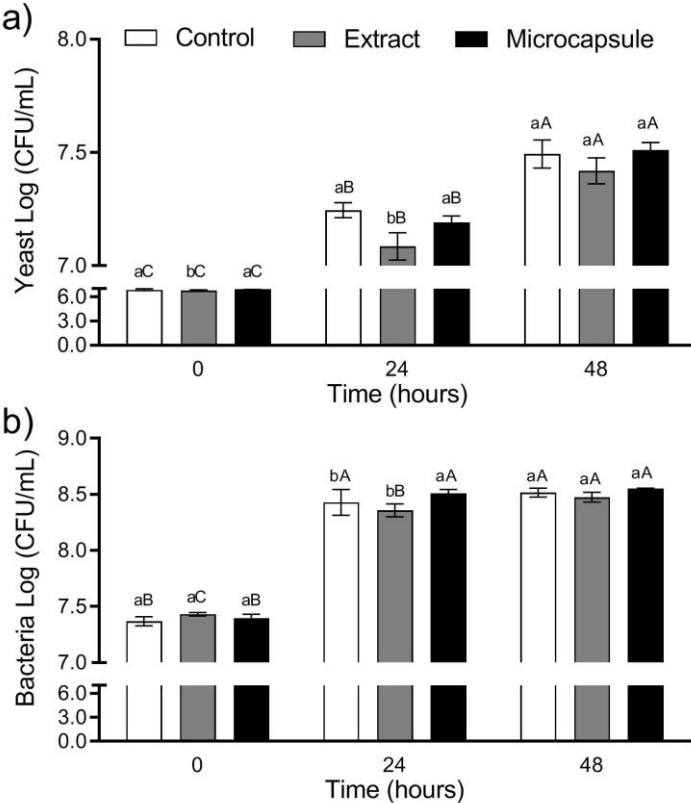

**Figure 2.** Microbial population during fermentation. (**a**) Yeast population. (**b**) L.A.B. population. Different lowercase letters denote differences ($p < 0.05$) among assays simultaneously. Different capital letters denote differences ($p < 0.05$) in the same assay at different times according to the Scott–Knott test. [Non-dairy matrix (N-DM): 1 Control—consists only of the fermented beverage (N-DM); 2 Extract—beverage (N-DM) with propolis extract (DPE); 3 Microcapsule—beverage (N-DM) with microencapsulated propolis (D.M.P.)].

Pathogenic microorganisms such as enterobacteria and filamentous fungi were not identified at the beginning or the end of the fermentation process (data not shown).

Figure 3 shows the pH profile during fermentation. The acidification of the beverages was evaluated at 0, 48 h, 14 days, and 28 days of storage. In general, all tests showed a reduction in pH during fermentation until the end of storage. Initially, the beverages had an average pH of 6.35. With 48 h of fermentation, the treatments showed a reduction ($p < 0.05$) of 23.63% for the control (4.85) and 22.05% for the D.M.P. (4.95) and DPE (4.95) assays. On the 14th day of storage, the pH values were slightly reduced ($p < 0.05$). The D.M.P. test showed a decrease of 12.13% with values around 4.35, while the control and DPE falls were 10.31% (4.35) and 7.08% (4.60), respectively. Observing the data from the 14th day of storage and comparing it with the 28th day, it is possible to notice that the pH values remained stable for all treatments. At the end of storage, the control and D.M.P. were the lowest ($p < 0.05$) pH values (4.15 and 4.30, respectively) when compared to the DPE (4.60). A low pH is necessary for fermented food products to prevent the growth of food pathogens and deteriorators microorganisms. The pHs recorded in the beverages at the end of storage were equal to or lower than a pH of 4.6, the maximum limit for fermented food products [52]. Therefore, *D. hansenii* CCMA 1761 and *L. plantarum* CCMA 0743 are good starter cultures for fermented probiotic beverages.

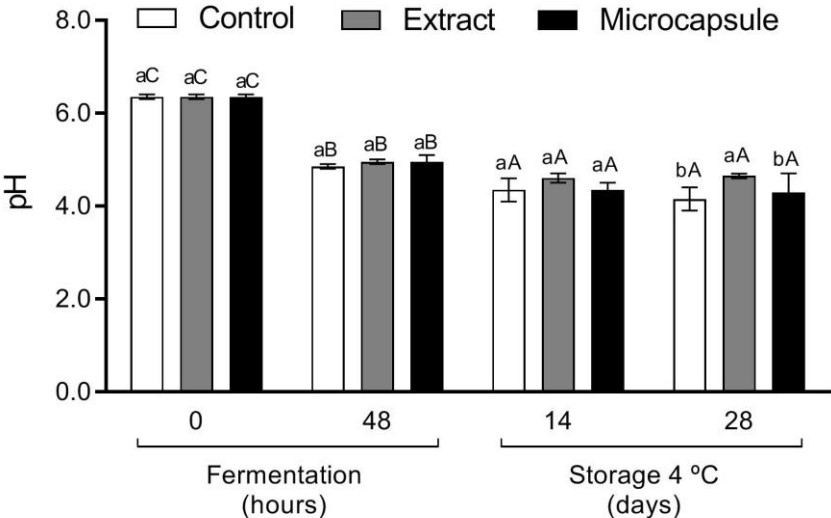

**Figure 3.** Reduction in pH during the fermentation process. Different lowercase letters denote differences ($p < 0.05$) among assays simultaneously. Different capital letters denote differences ($p < 0.05$) in the same assay at different times according to the Scott–Knott test. [Non-dairy matrix (N-DM): 1 Control—consists only of the fermented beverage (N-DM); 2 Extract—beverage (N-DM) with propolis extract (DPE); 3 Microcapsule—beverage (N-DM) with microencapsulated propolis (D.M.P.)].

### 3.3. Viability during Storage

The biggest challenge for a probiotic product's effectiveness is maintaining the microorganism's viability since the number of cells is important in achieving health benefits [53]. The microbial viability was analyzed after 7, 14, 21, and 28 days of storage at 4 °C; the data are shown in Figure 4. All treatments showed a slight reduction in the viable count of microorganisms during storage. For *D. hansenii*, a 3–4% reduction in viability was observed, not differing between treatments, at 28 days of storage (Figure 4a). Regarding *L. plantarum*, the observed reduction was also around 3% for all treatments (Figure 4b). It was noticed that at all storage times (7 to 28 days), the D.M.P. treatment showed higher count values. At 28 days, the D.M.P. presented a count of 8.22 log CFU/mL, significantly higher than the control (8.18 log CFU/mL) and DPE (8.14 log CFU/mL).

For a food to be considered probiotic, the product to be consumed must have a concentration of at least 6 log CFU/mL. This ensures enough cells are ingested to remain viable during gastrointestinal transit and to affect the host [4]. Although reductions in counts were observed, the results show that both starter cultures were able to maintain the population above 6 log CFU/mL in combination with propolis at the end of refrigerated storage at 4 °C, remaining above 7 log CFU/ mL for yeast and 8 log CFU/mL for bacteria.

### 3.4. Consumption of Sugars and Metabolites Produced

Carbohydrates, organic acids, and alcohols at 0 h, 24 h after adding propolis, and 28 days of storage of the plant beverage are shown in Table 1. Fructose was the main carbohydrate detected, ranging from 0.91 g/L to 0.97 g/L, followed by glucose around 0.36 g/L. The glucose and fructose contents were practically all consumed during the fermentation process, representing, on average, 95.03% and 94.15%, respectively, of the total carbohydrate consumption. At the end of fermentation, sucrose showed a consumption of 38.12%, and after storage, it showed residual concentrations of around 0.1 g/L. The treatments did not differ regarding carbohydrate consumption during fermentation and storage. The higher consumption of glucose and fructose is because they are monosaccharides formed by chains in a single bond, while sucrose is a disaccharide (glucose + fructose) joined by glycosidic bonds; therefore, it is more challenging to break [54]. In some bacteria, the genes that make up the sucrose catabolic operons are expressed only when sucrose is present and when other preferred carbon sources are depleted [55], which may explain the

lower percentage of consumption of this carbohydrate. In this case, the order of preference should be glucose ≥ fructose > sucrose. Other studies also observed similar behaviors in the consumption of carbohydrates [36,56].

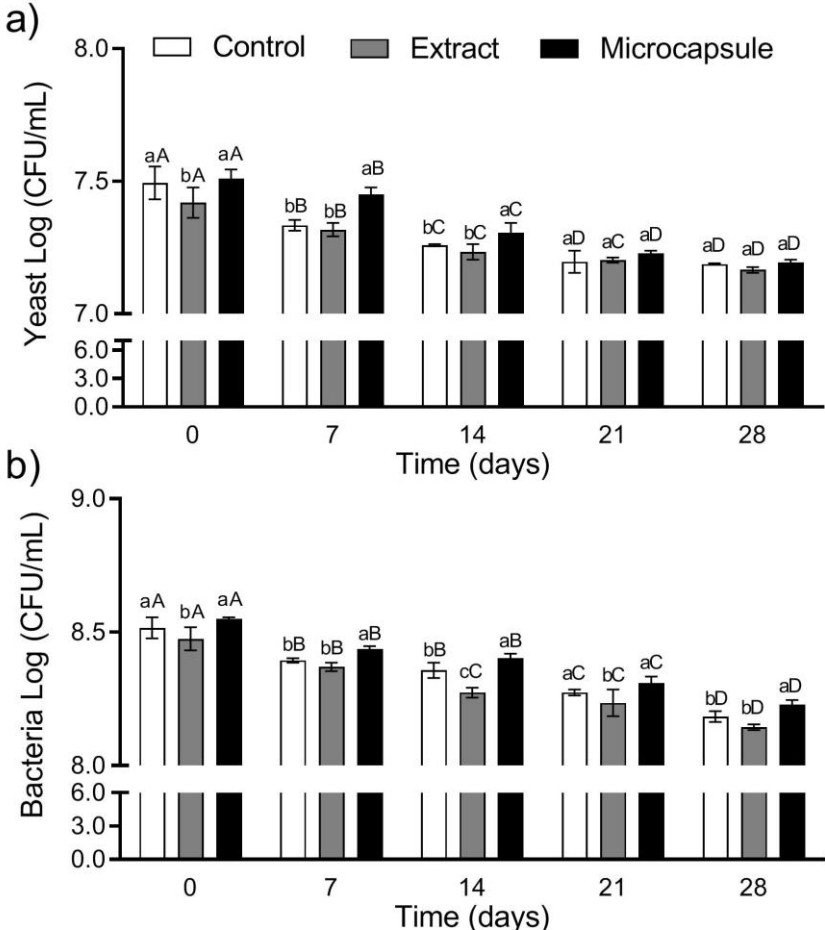

**Figure 4.** Microbial population during 28 days of storage at 4 °C. (**a**) Yeast population. (**b**) L.A.B. population. Different lowercase letters denote differences ($p < 0.05$) among assays during the same time. Different capital letters denote differences ($p < 0.05$) in the same assay at different times according to the Scott–Knott test. [Non-dairy matrix (N-DM): 1 Control—consists only of the fermented beverage (N-DM); 2 Extract—beverage (N-DM) with propolis extract (DPE); 3 Microcapsule—beverage (N-DM) with microencapsulated propolis (D.M.P.)].

Regarding organic acid concentrations, malic acid was present at the beginning of fermentation, around 0.94 g/L, but only residual concentrations were observed at the end of storage. Part of this organic acid was probably converted into lactic acid through malolactic fermentation, defined as the bioconversion of L-malic acid into L-lactic acid with the production of $CO_2$ [57]. Overall, lactic acid production increased toward the end of storage. The test with microencapsulated propolis showed the highest ($p < 0.05$) lactic acid concentration at the end of fermentation and at the end of storage, 1.25 g/L and 1.11 g/L, respectively. The replacement of L-malic dicarboxylic acid, characterized by a harsh taste, by L-lactic monocarboxylic acid, characterized by milder flavors, results in the deacidification of the beverage with a concomitant change in its gustatory and olfactory perception [58].

**Table 1.** Concentration of carbohydrates, organic acids, and alcohols in non-dairy beverages plus propolis (Figures S1 and S2; Table S1).

| Fermentation Stages | Samples | Concentration of Compounds (g/L) | | | | | | |
|---|---|---|---|---|---|---|---|---|
| | | Sucrose | Glucose | Fructose | Malic Acid | Lactic Acid | Acetic Acid | Ethanol |
| **0 h** | Control | $0.33 \pm 0.06$ [Aa] | $0.34 \pm 0.23$ [Aa] | $0.91 \pm 0.12$ [Aa] | $0.93 \pm 0.06$ [Aa] | ND [Cd] | ND [Cc] | ND [Bc] |
| | Extract | $0.32 \pm 0.10$ [Aa] | $0.36 \pm 0.22$ [Aa] | $0.93 \pm 0.37$ [Aa] | $0.91 \pm 0.02$ [Aa] | ND [Cd] | ND [Cc] | ND [Bc] |
| | Microcapsules | $0.32 \pm 0.08$ [Aa] | $0.36 \pm 0.22$ [Aa] | $0.97 \pm 0.35$ [Aa] | $0.94 \pm 0.06$ [Aa] | ND [Cd] | ND [Bc] | ND [Bc] |
| **24 h (after the addition of propolis)** | Control | $0.20 \pm 0.01$ [Ba] | $0.01 \pm 0.01$ [Ba] | $0.06 \pm 0.01$ [Ba] | $0.25 \pm 0.06$ [Bb] | $0.60 \pm 0.06$ [Bb] | $0.09 \pm 0.00$ [Ba] | $1.40 \pm 0.66$ [Aa] |
| | Extract | $0.19 \pm 0.03$ [Ba] | $0.003 \pm 0.00$ [Ba] | $0.05 \pm 0.01$ [Ba] | $0.18 \pm 0.08$ [Bb] | $0.54 \pm 0.01$ [Bb] | $0.06 \pm 0.00$ [Bb] | $1.50 \pm 0.14$ [Aa] |
| | Microcapsules | $0.21 \pm 0.17$ [Ba] | $0.04 \pm 0.01$ [Ba] | $0.05 \pm 0.01$ [Ba] | $0.41 \pm 0.01$ [Ba] | $1.25 \pm 0.06$ [Aa] | $0.11 \pm 0.00$ [Aa] | $1.11 \pm 0.01$ [Aa] |
| **28 days of storage** | Control | $0.10 \pm 0.02$ [Ba] | $0.10 \pm 0.03$ [Ba] | $0.07 \pm 0.01$ [Ba] | $0.27 \pm 0.13$ [Bb] | $0.82 \pm 0.05$ [Ab] | $0.16 \pm 0.05$ [Ab] | $1.43 \pm 0.55$ [Aa] |
| | Extract | $0.12 \pm 0.09$ [Ba] | $0.001 \pm 0.00$ [Ba] | $0.05 \pm 0.01$ [Ba] | $0.21 \pm 0.07$ [Bb] | $0.67 \pm 0.05$ [Ac] | $0.16 \pm 0.02$ [Aa] | $1.62 \pm 0.07$ [Aa] |
| | Microcapsules | $0.11 \pm 0.08$ [Ba] | $0.10 \pm 0.08$ [Ba] | $0.04 \pm 0.04$ [Ba] | $0.37 \pm 0.09$ [Ba] | $1.11 \pm 0.07$ [Ba] | $0.11 \pm 0.01$ [Ab] | $1.13 \pm 0.06$ [Ab] |

ND = not detected. Mean ± standard deviation. Different lowercase letters in the same column denote differences ($p < 0.05$) between treatments at the same time. Different capital letters in the same column denote differences ($p < 0.05$) in the same treatment at different times according to the Scott–Knott test. Compound retention time: sucrose, 10.269; glucose, 12.193; fructose, 14,902; malic acid, 12.718; lactic acid, 16.193; acetic acid, 18.555; ethanol, 18.883. [Non-dairy matrix (N-DM): **1 Control**—consists only of the fermented beverage (N-DM); **2 Extract**—beverage (N-DM) with propolis extract (DPE); **3 Microcapsule**—beverage (N-DM) with microencapsulated propolis (D.M.P.)].

Furthermore, removing L-malic acid, a potential carbon source for some spoilage yeasts [59], increases the stability of the final product. Acetic acid was detected at the end of fermentation and remained until the end of storage in concentrations ranging from 0.06 g/L to 0.16 g/L. In addition to lactic acid production, L. plantarum can also produce acetic acid and/or ethanol as products due to the facultative heterofermentative metabolism expressed under restrictive glucose conditions [60,61]. Zalán et al. [62] confirmed that some strains of Lactobacillus could change their fermentative profile depending on the composition of the medium.

For alcohols, only ethanol was detected at the end of fermentation. There was no significant difference ($p > 0.05$) in ethanol production at the end of the fermentation process. At the end of the 28 days of refrigerated storage, the propolis microcapsule test showed the lowest ($p < 0.05$) alcohol content, around 1.13 g/L. Ethanol values for the control and trial containing propolis extract were between 1.43 g/L and 1.62 g/L, respectively. According to Ignat et al. [63], to be considered a non-alcoholic beverage, the alcohol content must be below 5 g/L (0.5% *v/v*). All tests showed ethanol values below 2 g/L, being characterized as non-alcoholic beverages.

### 3.5. Volatile Compounds

Volatile compounds are responsible for beverages' aromas and unique flavor characteristics [64]. The beverage compounds were analyzed at the beginning of fermentation (substrate), at the end of fermentation, and at the end of 28 days of storage at 4 °C (Table 2). In total, 39 compounds were identified, including acids, alcohols, aldehydes, alkanes, alkenes, esters, ethers, phenols, terpenes, and others. Compounds with pleasant aromas such as floral, fruity, fresh, and sweet, among others, were present at the end of fermentation and storage. In general, volatile concentrations increased after fermentation.

Table 2. Volatile compounds identified by GC-MS analysis at the beginning (substrate), after fermentation (48 h, 24 h fermentation + 24h after adding propolis), and at the end of storage (28 days) (Figure S3).

| Chemical Class | Volatile Compounds | Ret. Time | Ret. Index | Sensory Perception | Concentration (µg/mL) | | | | | | |
| --- | --- | --- | --- | --- | --- | --- | --- | --- | --- | --- | --- |
| | | | | | Substrate | After Fermentation | | | After Storage | | |
| | | | | | | Control | Extract | Microcapsule | Control | Extract | Microcapsule |
| Acids | Acetic acid | 11.762 | 576 | Pungent, acid, cheese, vinegar [a] | - | 516.7 | 608.2 | 513.1 | 966.9 | 602.5 | 351.3 |
| | Octanoic acid | 19.032 | 1173 | - | 13.9 | 56.0 | - | 56.6 | 66.7 | - | 50.9 |
| | n-Decanoic acid | 21.049 | 1372 | - | 11.2 | 71.8 | 85.6 | 70.3 | 51.5 | 47.4 | 42.2 |
| | Benzoic acid | 22.708 | 1150 | Pungent, sour [b] | 11.9 | 7.1 | 105.9 | 2.7 | 6.7 | 26.6 | - |
| | 9,12-Octadecadienoic acid | 23.155 | 2183 | - | 4.1 | 1.8 | - | 4.0 | - | - | - |
| | n-Hexadecanoic acid | 27.516 | 1968 | Practically odorless and smooth flavor [a] | - | 32.3 | 95.1 | 57.9 | 31.1 | 21.3 | 21.2 |
| Alcohols | 1-Butanol, 3-methyl | 7.639 | 697 | Fuel oil, whiskey characteristic, pungent odor [c] | - | 94.8 | 34.6 | 131.7 | 78.5 | 23.8 | 123.6 |
| | 1-Hexanol | 10.444 | 860 | Flavoring ingredient: fruity odor and aromatic flavor [c] | 243.6 | 451.8 | 301.4 | 604.6 | 359.7 | 205.3 | 488.2 |
| | 1-Heptanol | 12.036 | 960 | Scented, woody, heavy, oily, weak, aromatic, greasy odor, and a spicy taste [c] | - | 36.6 | - | 52.1 | 32.0 | - | 31.6 |
| | 1,6-Octadien-3-ol, 3,7-dimethyl | 13.178 | 1082 | Coriander, floral, lavender, lemon, rose [b] | - | - | 48.8 | - | - | 33.9 | - |
| | 1-Octanol | 13.452 | 1059 | Fresh orange-pink odor, which is quite sweet [c] | 32.3 | 44.7 | - | 70.4 | 35.4 | - | 48.8 |
| | 1-Nonanol | 14.744 | 1159 | Fresh, orange, pink [c] | 31.6 | 47.0 | 50.6 | 68.9 | 41.4 | 34.2 | 49.1 |
| | L-.alpha.-Terpineol | 15.292 | 1143 | - | - | - | 90.5 | - | - | 67.8 | - |
| | Benzyl alcohol | 17.173 | 1036 | Sweet, floral [d] | 69.4 | 183.3 | - | 170.7 | 164.8 | - | 137.7 |
| | 1,6,10-Dodecatrien-3-ol, 3,7,11-trimethyl | 18.749 | 1564 | Pine [b] | - | - | 201.0 | - | - | 153.9 | - |
| Aldehydes | 2-Heptenal | 9.847 | 913 | Milky, green, greasy, and oily [f] | 43.3 | 28.8 | - | 57.1 | 9.3 | - | 26.8 |
| | Benzaldehyde | 12.753 | 982 | Pungent [e] | 36.2 | 100.3 | - | 103.9 | 71.4 | - | 70.2 |
| | Benzaldehyde, 4-(1-methylethyl) | 16.620 | 1230 | Acid, green, herb [b] | - | 42.3 | - | 56.7 | 50.1 | - | 110.9 |

**Table 2.** *Cont.*

| Chemical Class | Volatile Compounds | Ret. Time | Ret. Index | Sensory Perception | Concentration (µg/mL) | | | | | | |
|---|---|---|---|---|---|---|---|---|---|---|---|
| | | | | | Substrate | After Fermentation | | | After Storage | | |
| | | | | | | Control | Extract | Microcapsule | Control | Extract | Microcapsule |
| **Alkanes** | Nonane, 5-(2-methylpropyl) | 8.958 | 1185 | - | 42.4 | 6.6 | 21.3 | 6.6 | 22.6 | 16.7 | - |
| | Tetradecane | 11.440 | 1413 | - | 101.9 | - | - | - | - | - | 71.2 |
| | Heptadecane | 12.282 | 1711 | - | - | 20.3 | 33.5 | - | 1- | 26.0 | - |
| | Dodecane, 4,6-dimethyl | 12.303 | 1285 | - | - | - | - | 46.2 | 24.9 | - | 4 |
| | Hexadecane | 14.138 | 1612 | - | 32.2 | - | 37.8 | - | - | 37.8 | 24.7 |
| **Alkenes** | Copaene | 12.437 | 1221 | Wood, spice [d] | - | - | 263.3 | - | - | 165.7 | - |
| | Myrtenol | 16.332 | 1191 | Mint [b] | - | 119.1 | 103.4 | 189.6 | 107.8 | 89.3 | 142.1 |
| **Esters** | Octanoic acid, ethyl ester | 11.565 | 1883 | Fruit [d] | - | 94.0 | 75.1 | 104.1 | 81.5 | 77.7 | - |
| | Benzenepropanoic acid, ethyl ester | 17.241 | 1359 | Flower [d] | - | - | 8191.6 | - | - | 6825.3 | - |
| | Hexadecanoic acid, ethyl ester | 20.843 | 1978 | Wax [b,d] | - | - | 67.9 | - | - | 40.5 | - |
| | Ethyl oleate | 22.763 | 2185 | Dairy [b] | - | - | 19.6 | - | - | 7.3 | - |
| | Linoleic acid ethyl ester | 23.164 | 2193 | - | - | 1.9 | 41.6 | - | 3.8 | 16.0 | 2.1 |
| **Ethers** | Verbenyl, ethyl ether | 10.773 | 1184 | - | - | - | 32.6 | - | - | 24.2 | - |
| | Estragole | 11.691 | 1172 | Licorice [b,d] | 21.3 | 2.6 | - | 13.7 | 2.0 | - | 16.5 |
| | Methyleugenol | 18.601 | 1361 | Burnt, clove, spice [b] | - | - | 601.7 | 99.8 | - | 480.1 | 191.4 |
| **Phenol** | Phenol, 2,6-dimethoxy-4-(2-propenyl) | 23.461 | 1581 | Sweet, flower [d] | - | - | 34.4 | 3.2 | - | - | 5.2 |
| **Terpenes** | .beta.-Bisabolene | 15.446 | 1500 | Flower [b] | 14.1 | 17.9 | 136.4 | 23.2 | 14.2 | 89.4 | 16.6 |

**Table 2.** *Cont.*

| Chemical Class | Volatile Compounds | Ret. Time | Ret. Index | Sensory Perception | Concentration (µg/mL) | | | | | | |
|---|---|---|---|---|---|---|---|---|---|---|---|
| | | | | | Substrate | After Fermentation | | | After Storage | | |
| | | | | | | Control | Extract | Microcapsule | Control | Extract | Microcapsule |
| **Others** | Naphthalene, 1,2,3,5,6,8a-hexahydro-4,7-dimethyl-1-(1-methylethyl)-, (1S-cis) | 15.848 | 1469 | - | - | - | 95.0 | - | - | 65.4 | - |
| | Benzene, 1-(1,5-dimethyl-4-hexenyl)-4-methyl | 15.915 | 1524 | Herb [d] | 10.5 | - | - | - | - | - | - |
| | Benzene, 1,2,3-trimethoxy-5-(2-propenyl) | 22.272 | 1550 | - | - | - | 73.0 | 10.4 | - | 40.1 | 21.5 |

Values are expressed as the mean concentration (µg/mL). Sensory perception not found / compound not detected. Sensory perceptions were mentioned in a, [65]; b, [66]; c, [67]; d, [68]; e, [69]; and f, [70]. [Non-dairy matrix (N-DM)**: 1 Control**—consists only of the fermented beverage (N-DM); **2 Extract**—beverage (N-DM) with propolis extract (DPE); **3 Microcapsule**—beverage (N-DM) with microencapsulated propolis (D.M.P.)].

Alcohols were the most abundant volatile group with nine compounds. In addition to ethanol, many alcohols are also produced by yeast. These higher alcohols (fusel alcohols) are secondary metabolites formed by the Ehrlich pathway [71]. Among alcohols, 1-nonanol, 1-hexanol and 1-butanol, 3-methyl were the only ones identified in all treatments, the tests with propolis microcapsules having the highest concentrations. These compounds are associated with fresh, orange, rose, and fruity aromas. Amyl alcohols such as 1-butanol, 3-methyl are formed during fermentation by deamination and decarboxylation of isoleucine [72,73]. The volatiles 1,6-octadien-3-ol, 3,7-dimethyl, L-.alpha.-terpineol and 1,6,10-dodecatrien-3-ol, 3,7,11-trimethyl were identified only in treatments containing propolis extract, the latter being the most abundant compound (201 μg/mL at the end of fermentation and 153.9 μg/mL at the end of storage) with a pine aroma. Benzyl alcohol was not detected only for treatments with propolis extract. The presence of this compound can contribute to floral and sweet notes [74–76], which can be considered a positive characteristic for beverages.

Regarding acids, acetic acid and n-hexadecanoic acid were detected only after fermentation and remained until the end of storage. The production of n-hexadecanoic acid by yeasts is well reported in the literature [62,70,71], and its concentration in the beverages ranged from 351.3 μg/mL to 966.9 μg/mL. Acetic acid is a short-chain fatty acid (SCFA). SCFAs are small organic monocarboxylic acids with different chain lengths, ranging from two to six carbon atoms. They are products of the fermentation of dietary polysaccharides, including fiber and resistant starch [75–77]. Evidence shows that SCFAs can be relevant in managing metabolic diseases, including obesity and diabetes [78].

Furthermore, SCFAs can also limit the onset of inflammatory processes by acting as signaling molecules, reducing the production of proinflammatory cytokines [79]. Long-chain fatty acids (LCFA) such as 9,12-octadecadienoic acid (linoleic acid) were identified only in the test with microencapsulated propolis at the end fermentation in small concentrations (ranging from 1.8 μg/mL to 4.0 μg/mL). The intake of monounsaturated fatty acids may reduce the risk factors for heart disease and stroke by lowering blood cholesterol and triglyceride levels [80].

Aldehydes are obtained as a result of alcohol oxidation during fermentation [66]; in addition, they can be converted into other compounds during beverage storage [81]. The test containing microencapsulated propolis was the only one that showed the compound benzaldehyde, 4-(1-methylethyl). Its concentrations ranged from 56.7 μg/mL at the end of fermentation to 110.9 μg/mL at the end of storage.

Yeasts can catabolize amino acids through the Ehrlich pathway or use the acetyl-CoA produced in glycolysis to generate esters [82,83]. Esters with floral, fruity, and dairy aromas were identified during fermentation. The test containing propolis extract was the only one that showed all the esters detected, highlighting benzenepropanoic acid and ethyl ester as floral aromas that showed high concentrations both at the end of fermentation and at the end of storage (8191.6 μg/mL and 6825.3 μg/mL, respectively). However, for the test with microencapsulated propolis, only two compounds were identified: octanoic acid ethyl ester and linoleic acid ethyl ester, the latter being in very low concentrations (2.1 μg/mL).

Compounds such as copaene, beta-bisabolene, alkenes, and terpenes were also identified. These compounds are characteristic of propolis. De Oliveira et al. [78], while evaluating the chemical composition of the volatile fraction of seven samples of propolis collected in Northeast Pará, also observed the presence of these compounds, but in lower concentrations.

Principal component analysis (P.C.A.) correlated the functional groups identified with the different assays at the end of fermentation (Figure 5a). P.C.A. explained 88.88% of the total variability in two dimensions (64.09% for the first and 24.79% for the second dimension). In the positive quadrant of F2, the test with propolis extract was correlated mainly with esters since these represented more than 50% of the identified functional groups (Figure 5b). However, on the positive side of F1, the tests with microencapsulated propolis and the control (fermented matrix) were grouped and correlated mainly with

acids and alcohols since the percentage of the two groups represents more than 60% in both treatments.

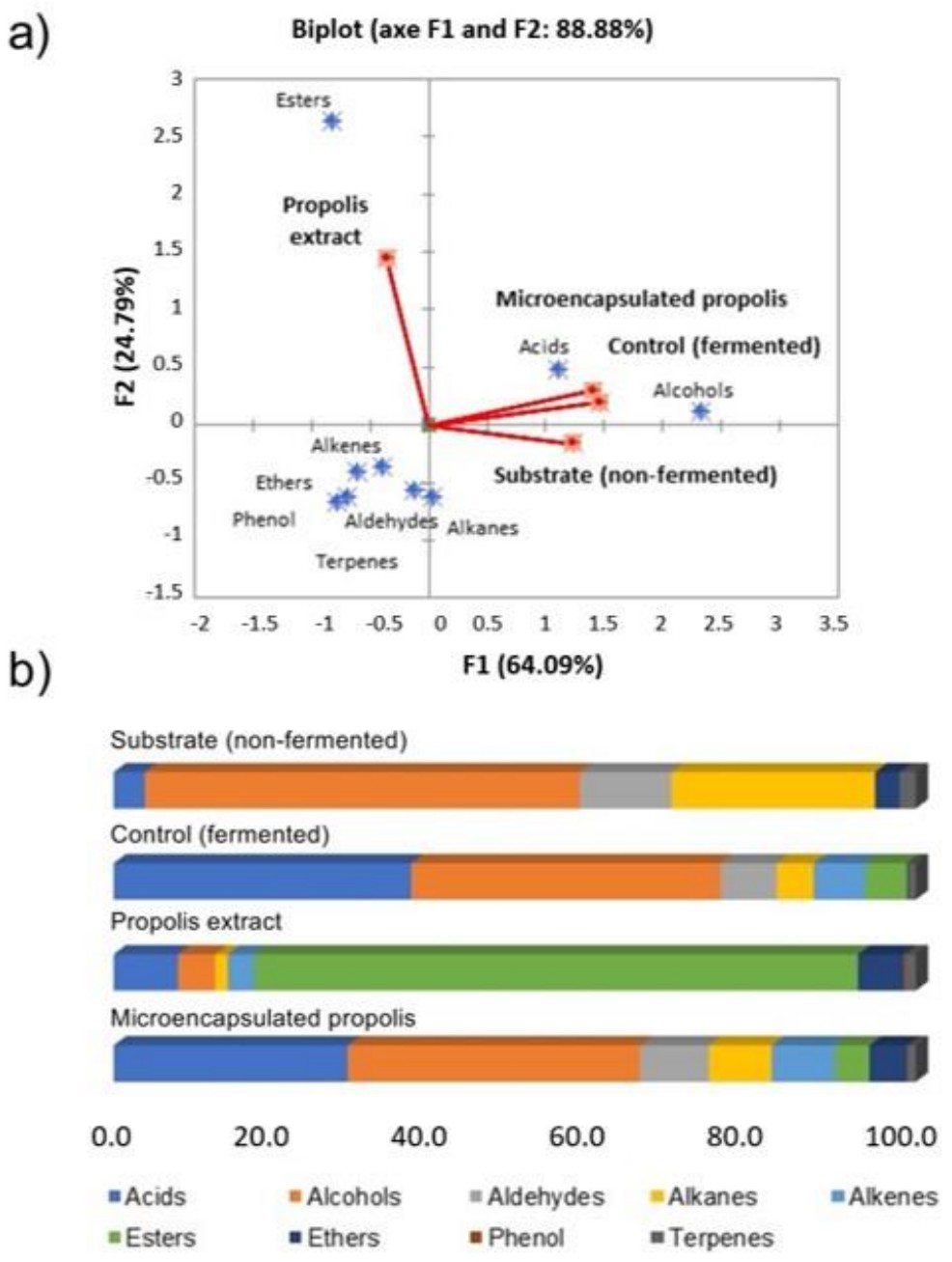

**Figure 5.** (**a**) Principal component analysis of chemical groups and assays. (**b**) Area percentage of volatile groups identified by GC-MS. [Non-dairy matrix (N-DM)**: 1 Control**—consists only of the fermented beverage (N-DM); **2 Extract**—beverage (N-DM) with propolis extract (DPE); **3 Microcapsule**—beverage (N-DM) with microencapsulated propolis (D.M.P.)].

*3.6. Antioxidant Activity and Phenolics*

The health benefits of herbal products are often related to the antioxidant activity and their phenolic compounds [84]. Antioxidants are oxidizing compounds that inhibit, prevent, or delay cell damage caused by free radicals and unstable molecules present in human cells [84,85]. Figure 6 shows the results of the antioxidant activity by ABTS and the phosphomolybdenum complex (P.C.M.) method and the total phenolic content. For the ABTS method (Figure 6a), the antioxidant activity increased ($p < 0.05$) only for the DPE assay, showing values of 3.56 µMol T.E./mL at the end of fermentation and 3.22 µMol T.E./mL

at the end of storage. Regarding P.C.M. (Figure 6b), the fermentation process reduced the beverage's antioxidant activity. A previous study demonstrated that some Lactobacillus species could use antioxidant compounds as substrates during fermentation, resulting in reduced antioxidant potential [86]. Even with the reduction, the DPE assay showed a higher ($p < 0.05$) antioxidant activity at the end of fermentation (10.74 mg/mL of ascorbic acid) and at the end of storage (10.72 mg/mL of ascorbic acid). According to Martinez et al. [87], the antioxidant activity may depend on the number of the total phenolic compounds or compounds with antioxidant capacities. Figure 6c shows the phenolic values. It can be noted that the DPE assay showed the highest ($p < 0.05$) concentrations of phenolics, with 6.79 mg G.A.E./mL at the end of fermentation and 6.76 mg G.A.E./mL at the end of storage. According to Dani et al. [88], higher concentrations of phenolics may imply a greater antioxidant activity, as observed for the DPE test. The other treatments showed no significant difference ($p > 0.05$) over time. The phenolic compounds present in propolis may contribute to the functional properties of the beverage, including antioxidant, antimicrobial, antiviral, anti-inflammatory, antifungal, and wound healing [89].

### 3.7. Viscosity

Rheology is the science that studies the deformation of matter (food), varying in the transformation from food processing to the final product [90], with viscosity being one of the most important parameters in food processing for the design of equipment, determination of the functionality of the ingredient, quality control of the intermediate or final product, and evaluation of the valuable life [91].

The viscosity values of the beverages are shown in Table 3. There was no significant difference ($p > 0.05$) in the viscosity values after fermentation for the propolis extract test and the control. Only the treatment containing microencapsulated propolis showed an increased viscosity ($p < 0.05$). The authors suggest that this increase may be related to the addition of solids (propolis microcapsules) in the beverage. One of the most common texture/viscosity issues in plant-based fermented products is phase separation, and this significantly influences consistency, final product appearance, sensory perception, and acceptability of foods during consumption [92,93].

**Table 3.** Beverage viscosity at the beginning and end of fermentation.

| Time | Essay | | |
|---|---|---|---|
| | Microcapsule | Extract | Control |
| **Start of fermentation** | 3.24 ± 0.09 [aB] | 3.20 ± 0.01 [aA] | 3.22 ± 0.01 [aA] |
| **End of fermentation** | 3.38 ± 0.01 [aA] | 3.23 ± 0.06 [bA] | 3.21 ± 0.06 [bA] |

Different lowercase letters denote differences ($p < 0.05$) between treatments during the same time. Different capital letters denote differences ($p < 0.05$) in the same treatment at different times according to the Scott–Knott test. [Non-dairy matrix (N-DM): 1 Control—consists only of the fermented beverage (N-DM); 2 Extract—beverage (N-DM) with propolis extract (DPE); 3 Microcapsule—beverage (N-DM) with microencapsulated propolis (D.M.P.)].

Regarding the behavior of the fluids, the rheograms show that, for all treatments, the shear stress was directly proportional to the shear rate, indicating a Newtonian behavior (Figure 7).

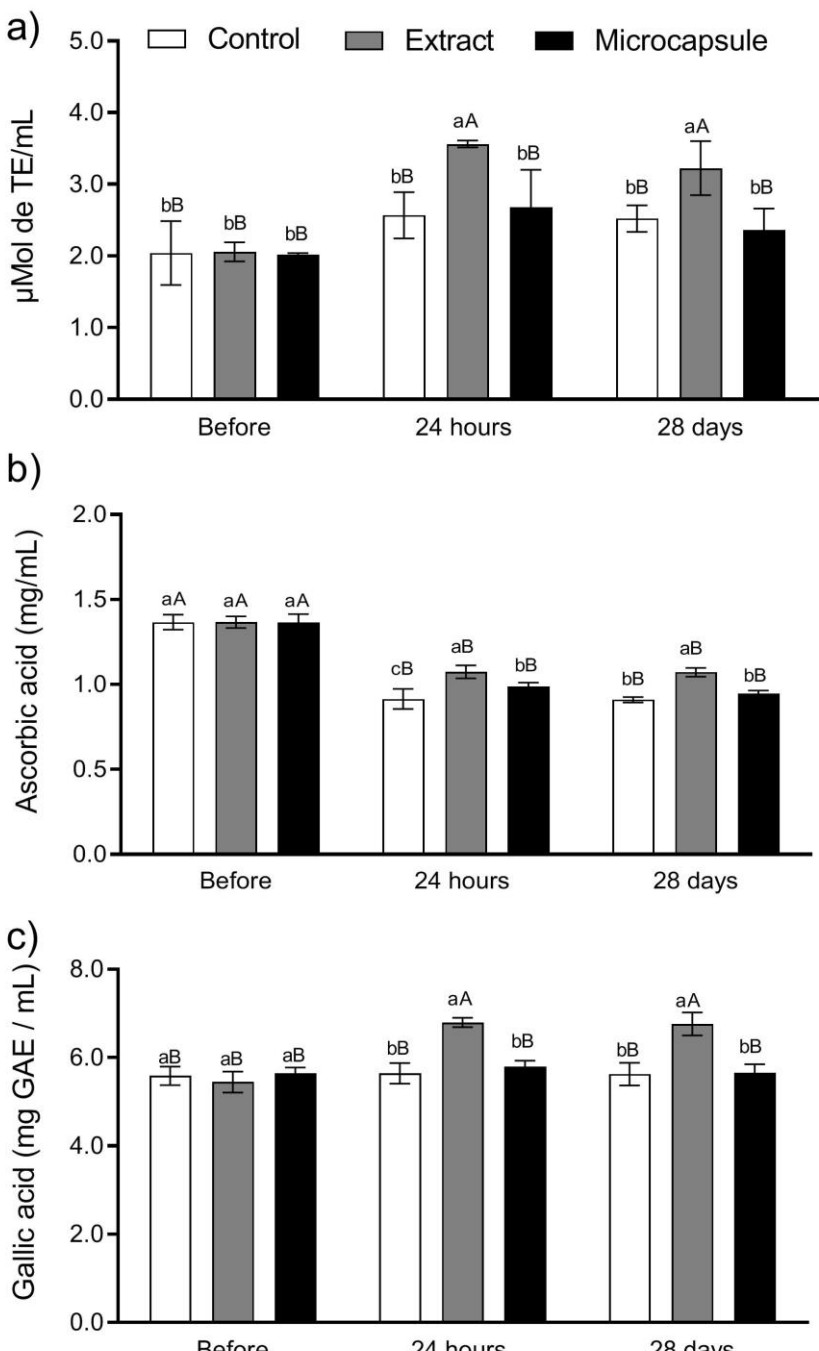

**Figure 6.** Antioxidant activity by ABTS radical reduction, the phosphomolybdenum complex method, and the total phenolic content. (**a**) Antioxidant activity by the ABTS radical reduction method. (**b**) Antioxidant activity by the phosphomolybdenum complex method. (**c**) Total phenolic content. Different lowercase letters denote differences ($p < 0.05$) among assays simultaneously. Different capital letters denote the same assay's difference ($p < 0.05$) at different times according to the Scott–Knott test. [Non-dairy matrix (N-DM): 1 Control—consists only of the fermented beverage (N-DM); 2 Extract—beverage (N-DM) with propolis extract (DPE); 3 Microcapsule—beverage (N-DM) with microencapsulated propolis (D.M.P.)].

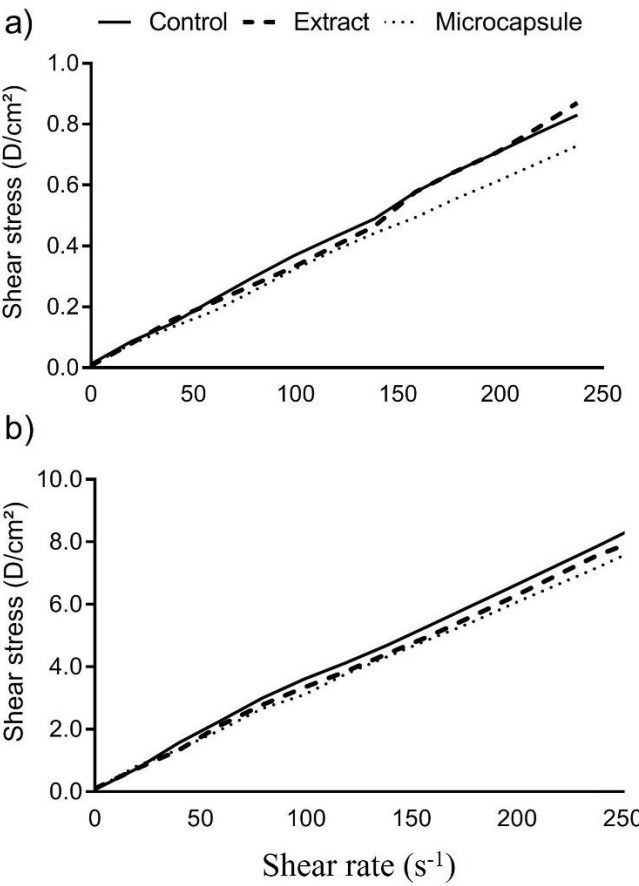

**Figure 7.** Rheograms of non-dairy beverages with propolis at 4 °C [Non-dairy matrix (N-DM): 1 Control—consists only of the fermented beverage (N-DM); 2 Extract—beverage (N-DM) with propolis extract (DPE); 3 Microcapsule—beverage (N-DM) with microencapsulated propolis (D.M.P.)].

*3.8. Sensory Analysis*

In total, 108 tasters participated in the sensory analysis. Only 33.33% of the panelists reported consuming non-dairy products, and only 5.55% said they were vegetarians. The consumer acceptance of the different non-dairy beverages with added propolis was evaluated, and the results are shown in Table 4. No sensory attribute showed a significant difference ($p > 0.05$) for the D.M.P. and control, demonstrating that adding microencapsulated propolis did not interfere with the sensory perception of the products. According to Maroof et al. [94], the microencapsulation process can mask bitter flavors and aromas such as propolis, which explains why tasters did not notice any sensory difference between the two treatments. The attribute "flavor" for the DPE test was the only one classified as "disliked moderately" with a score of 3.15. Although the volatile benzene propanoic acid, ethyl ester (flower aroma), was in very high concentrations, the strong aftertaste of propolis predominated. The intense and bitter taste of propolis extract may be related to compounds such as acetic and benzoic acid, characteristic of sensory perceptions such as vinegar and pungent aromas. Chon et al. [89], working with different concentrations of propolis in dairy products, observed that flavor attributes and overall acceptability tended to decrease as the concentration of propolis increased. The overall impression shows that the panelists neither liked nor disliked the beverages from the D.M.P. and control trials (scores 5.07 and 5.26, respectively). Regarding the purchase intention of the products, the D.M.P. and control were classified as "probably I would not buy" with scores of 2.60 and 2.67, respectively. On the other hand, the DPE test fell into the "definitely I would not buy it" category with a score of 1.80. Adding artificial sweeteners and flavorings is a possible option to make the beverage more pleasant to consumers, thus reducing the residual taste of propolis and making it possible to market the products.

**Table 4.** Acceptance of different fermented non-dairy beverages with potential probiotic co-cultures plus propolis.

| Treatments | Sensory Attributes | | | | |
|---|---|---|---|---|---|
| | Appearance | Texture | Flavor | Overall Impression | Buy Intention |
| **Microcapsule** | 7.00 [a] | 6.21 [a] | 4.42 [a] | 5.07 [a] | 2.60 [a] |
| **Extract** | 6.31 [b] | 5.93 [a] | 3.15 [b] | 4.03 [b] | 1.80 [b] |
| **Control** | 6.68 [ab] | 6.21 [a] | 4.64 [a] | 5.26 [a] | 2.67 [a] |

The scores for the consumer acceptance test are means $\pm$ S.D. According to Tukey's test, values with different letters are significantly different ($p < 0.05$). Acceptability was assessed using a nine-point structured hedonic scale ranging from 1 (disliked very much) to 9 (liked very much). [Non-dairy matrix (N-DM): 1 Control—consists only of the fermented beverage (N-DM); 2 Extract—beverage (N-DM) with propolis extract (DPE); 3 Microcapsule—beverage (N-DM) with microencapsulated propolis (D.M.P.)].

## 4. Conclusions

Non-dairy beverages based on oats, sunflower seeds, and almonds fermented by co-culturing with probiotic attributes with added Brazilian red propolis proved to be excellent functional beverages. The tested strains showed adequate growth and viability during fermentation with viable cells above $10^6$ log CFU/mL after 28 days of refrigerated storage, even in propolis. The beverages produced with propolis extract showed a higher antioxidant activity, phenolic concentration, and the presence of volatile esters. The test with microencapsulated propolis, on the other hand, was more associated with the presence of higher alcohols, higher concentrations of lactic and acetic acid (1.25 g/L and 0.11 g/L, respectively), and presented a better global impression score in the sensory analysis, not differing from the control. This proves that the microencapsulation process could mask propolis' intense characteristic flavors and odors. More studies to improve the acceptability and verification of health benefits should be conducted to improve the development of this new functional fermented beverage.

**Supplementary Materials:** The following supporting information can be downloaded at: https://www.mdpi.com/article/10.3390/fermentation9030234/s1, Figure S1. HPLC chromatograms; Figure S2. Calibration curve-HPLC; Table S1. Alkanes and their respective retention time and index; Figure S3. GC.

**Author Contributions:** I.F.: conceptualization, methodology, investigation, formal analysis, data curation, and writing original draft preparation. D.d.S.M. and K.T.M.-G.: writing, review and editing, visualization, and supervision. R.F.S.: supervision, writing—review and editing, project administration, and funding acquisition. R.C.d.C.A., C.O.d.S. and D.R.D.: review and editing. M.S.S., C.S.F.-T. and L.S.P.: preparation of the propolis microcapsules. All authors have read and agreed to the published version of the manuscript.

**Funding:** The authors are grateful to the "Coordenação de Aperfeiçoamento de Pessoal de Nível Superior-Brazil (Capes)-Code 001. Process No: 88887.682738/2022-00-UFBA".

**Institutional Review Board Statement:** Not applicable.

**Informed Consent Statement:** Not applicable.

**Data Availability Statement:** Data available upon request.

**Acknowledgments:** The authors thank the following Brazilian agencies: Conselho Nacional de Desenvolvimento Científico e Tecnológico (CNPq), Fundação de Amparo a Pesquisa do Estado de Minas Gerais (FAPEMIG), and Coordenação de Aperfeiçoamento de Pessoal de Nível Superior (CAPES).

**Conflicts of Interest:** The authors confirm that they have no conflicts of interest with respect to the work described in this manuscript.

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
