# Peer review of "Non-Lactic Probiotic Beverage Enriched with Microencapsulated Red Propolis: Microorganism Viability, Physicochemical Characteristics, and Sensory Perception"

_fermentation, doi:10.3390/fermentation9030234_

Round 1

Reviewer 1 Report

Dear authors, 

First of all congratulation for your huge work. I believe that this manuscript can be divided into more than one articles, but you had an holistic approach of your subject. My suggestion is that manuscript must be accepted after some minor corrections.

My comments are: 

In section 2.1 please add some more information about the red propolis and add information about the production (quantity, number of samples, area etc)

In lines 391-392: Seperate the legend of the table from the text 

In table 2: make it better to read (bigger columns, smaller letters etc) 

For all of the legends use smaller letters to differ them from the main text

I do not know how many figures and tables you can include in your manuscript, if you have more add the as supplemtary material

In conclusion, I believe that you have to add some more information, but it is your decision 

Author Response

Reviewer #1: To Authors:

Dear Reviewer

We made all corrections to the manuscript. Corrections are highlighted in yellow (and marked up using the “Track Changes”) in the attached manuscript. The authors thank the review. The corrections improved the manuscript. The manuscript has been corrected in terms of English writing. Supplementary materials S1-S4 have been added as a zip file.

We hope that the resubmitted paper meets the standards for publication in Fermentation.

_______________________

Dear authors, 

First of all, congratulation for your huge work. I believe that this manuscript can be divided into more than one articles, but you had a holistic approach of your subject. My suggestion is that manuscript must be accepted after some minor corrections.

My comments are: 

In section 2.1 please add some more information about the red propolis and add information about the production (quantity, number of samples, area etc).

Authors: Information on characteristics of red propolis was added in the manuscript.

In lines 391-392: Separate the legend of the table from the text.

Authors: Thank you for the suggestion, the correction was made.

In table 2: make it better to read (bigger columns, smaller letters etc).

Authors: Thank you for the suggestion, the correction was made.

For all of the legends use smaller letters to differ them from the main text.

Authors: Thank you for the suggestion, the correction was made.

I do not know how many figures and tables you can include in your manuscript, if you have more add the as supplemtary material.

Authors: Thank you for the suggestion. We have added some figures in supplemental material.

In conclusion, I believe that you have to add some more information, but it is your decision.

Authors: Thank you for the suggestion. The changes were made.

Reviewer 2 Report

The manuscript entitled “Non-lactic probiotic beverage enriched with microencapsulated red propolis: microorganism viability, physicochemical characteristics, and sensory perception” The present manuscript has been written well but I have following queries and suggestions?

In abstract author mention about HPLC and GCMS analyses but HPLC and GCMS results not presented, Author should include results obtained from HPLC and GCMS.

Add full name first for HPLC and GCMS after that you can use abbreviation

Introduction section lack of information about oats, sunflower seeds, almonds and chromatographic analysis using GCMS and HPLC.

Section 2.7. Analysis of organic acids, alcohols, and carbohydrates: Which standard used and their sample preparation not available? Solvent system not reported.

Chromatographic conditions should be recheck and rewrite.

Section 3.4. Consumption of sugars and metabolites produced: As per my understanding this section represent HPLC results?

How you determine the concentrations of Carbohydrates, organic acids, and alcohol. Which standard you used no details mention.

What was the concentration and retention time no information?

Also author should include HPLC chromatograms of standard and samples. Also include calibration curve and linearity.

Table represents HPLC results? Please add retention time of the compounds.

Table 2 data: Authos should include retention time of compounds and retention index alkanes (C7–C19).

Author should include GCMC chromatograms of samples.

Conclusion should be revised and highlight the GCMS and HPLC findings and their correlation.

Author Response

Reviewer #2: To Authors:

Dear Reviewer

We made all corrections to the manuscript. Corrections are highlighted in yellow (and marked up using the “Track Changes”) in the attached manuscript. The authors thank the review. The corrections improved the manuscript. The manuscript has been corrected in terms of English writing. Supplementary materials S1-S4 have been added as a zip file.

We hope that the resubmitted paper meets the standards for publication in Fermentation.

_______________________

The manuscript entitled “Non-lactic probiotic beverage enriched with microencapsulated red propolis: microorganism viability, physicochemical characteristics, and sensory perception” The present manuscript has been written well but I have following queries and suggestions.

In abstract author mention about HPLC and GCMS analyses but HPLC and GCMS results not presented, Author should include results obtained from HPLC and GCMS.

Authors: Thank you for the suggestion. The changes were made in abstract.

Add full name first for HPLC and GCMS after that you can use abbreviation.

Authors: Thank you for the suggestion. The change was made.

 Introduction section lack of information about oats, sunflower seeds, almonds and chromatographic analysis using GCMS and HPLC.

Authors: Thank you for the suggestion. The changes were made in introduction.

Section 2.7. Analysis of organic acids, alcohols, and carbohydrates: Which standard used and their sample preparation not available? Solvent system not reported. Chromatographic conditions should be recheck and rewrite.

Authors: Thank you for the suggestion. The sample preparation method and the solvent system were informed in topic 2.7.

All compounds identified had external standards of chromatographic degree as parameters. The external standards were:

Sucrose, glucose, malic acid, lactic acid, acetic acid and Ethanol - Sigma-Aldrich (Saint Louis, MO, United States).

Fructose – Merck (Darmstadt, Germany).

Section 3.4. Consumption of sugars and metabolites produced: As per my understanding this section represent HPLC results? How you determine the concentrations of Carbohydrates, organic acids, and alcohol. Which standard you used no details mention. What was the concentration and retention time no information?

Authors: Data have been added in Table 1.

Yes, this section represents HPLC results.

Compounds were identified based on the retention time of external standards and their concentrations were determined using a standard curve. The standards were at a concentration of 4 g/L.

The default retention times are:

Ethanol - 18.883/407041

Glucose – 12.193/793557

Fructose – 14.902/765960

Sucrose – 10.269/798696

Lactic acid – 16.193/4010961

Acetic acid – 18.555/ 4283777

Malic acid – 12.718/6414290

NOTE: retention time / area. The retention times may show a small variation of  ± 0.4.

All compounds identified had external standards of chromatographic degree as parameters. The external standards were:

Sucrose, glucose, malic acid, lactic acid, acetic acid and Ethanol - Sigma-Aldrich (Saint Louis, MO, United States).

Fructose – Merck (Darmstadt, Germany).

Also author should include HPLC chromatograms of standard and samples. Also include calibration curve and linearity.

Authors: Thank you for the suggestion. The HPLC chromatograms can be found in the "Supplementary Material": " S1 - HPLC_CHROMATOGRAMS" file. The standard curve is found in the " S2 - Calibration_curve_HPLC" file.

Table represents HPLC results? Please add retention time of the compounds.

Authors: Yes, Table 1 represents the HPLC results. The retention time of the compounds was included in the caption of Table 1.

Table 2 data: Authors should include retention time of compounds and retention index alkanes (C7–C19).

Authors: Thank you for the suggestion. The retention times and index of all compounds were entered in Table 2. Supplementary Table 1 (S3 - Supp_file_table_1) shows all alkanes with their respective retention times and index.

Author should include GCMS chromatograms of samples.

Authors: GC analysis chromatograms are in the “S4 - Supp_imagesGC” supplementary file.

Conclusion should be revised and highlight the GCMS and HPLC findings and their correlation.

Authors: Thank you for the suggestion. The changes were made.